# Delayed Diagnosis of Bilateral Neuroarthropathy: Serious Impact on the Development of Charcot’s Foot, a Case Report

**DOI:** 10.3390/medicina58121763

**Published:** 2022-11-30

**Authors:** Tatiana Benard, Corinne Lequint, Anne Christine Jugnet, Marie Bouly, Thomas Winther, Alfred Penfornis, Dured Dardari

**Affiliations:** 1Centre Hopitalier Sud Francilien, Diabetology Department, 91100 Corbeil-Essonnes, France; 2Paris-Sud Medical School, Paris-Saclay University, 91100 Corbeil-Essonnes, France; 3LBEPS, IRBA, Université d’Evry, Université Paris Saclay, 91025 Evry, France

**Keywords:** Charcot neuroarthropathy, multifocal active neuroarthropathy, delayed diagnosis

## Abstract

Charcot neuroarthropathy (CN) is a destructive complication of the joints in patients with diabetes and should be managed from the onset of the first symptoms to avoid joint deformity and the risk of amputating the affected joint. Here, we describe the case of a young 24-year-old patient living with type I diabetes who developed active bilateral CN in both tarsal joints. This case of neuroarthropathy was uncommon due to the bilateral presentation at the same level of the joint. Despite the patient consulting from the beginning of the symptoms, none of the physicians suspected or diagnosed CN, leading to a delay in management and the aggravation of bone destruction by CN. This highlights the importance of early management of CN with the need to refer people with suspected CN to specialised diabetic foot care centres.

## 1. Introduction

Charcot neuroarthropathy (CN) is a destructive complication of the joint, which occurs mostly in people with diabetes. Knowledge about the cause of this pathology remains at the theoretical and hypothetical stage, with the more recent being the neuro-bone-inflammatory theory described by Jeffcoate in 2005 [1]. He described CN as an increased inflammatory response to a trauma or lesion, inducing increased bone lysis. The typical clinical presentation of acute Charcot foot is a red swollen foot with a temperature difference of more than 2 °C compared with the unaffected foot. These symptoms may go unnoticed, as pain may be absent or disproportionate. CN usually evolves in two phases—(i) acute and (ii) chronic—and signs and symptoms from both phases may be mixed. The most common description in the literature is the Eichenholtz classification based on clinical and radiological signs [2]. According to this classification, stage 0 (CN insipidus) is characterised by mild inflammation, soft tissue oedema and normal X-rays but abnormal magnetic resonance imaging (MRI) showing signs of microfracture, bone marrow oedema and bone contusion. The recognition and management of CN at stage 0 could stop disease activity and prevent foot deformity [3]. In a recent series in which MRI was performed very early, 69% of patients with stage 0 CN recovered without deformities in contrast to only 7% of patients with delayed presentation at stage 1 [4]. Stage 1 is characterised by severe inflammation, soft tissue oedema, abnormal X-rays with macro-fractures and abnormal MRI showing signs of macro-fractures and bone marrow oedema. In this stage, bone resorption sets in with the presence of joint dislocation. In stage 2 CN, coalescence occurs with the end of bone resorption and the start of remodelling in addition to the healing of fractures and resorption of debris. Finally, stage 3 is marked by definitive bone remodelling with bone reconstruction followed by the onset of the chronic phase of CN, which is synonymous with the frequent appearance of ulcers because of significant changes to the arch of the foot.

Despite the importance of the early diagnosis of CN, our case describes the common phenomenon of delayed management of CN and highlights the erroneous assumptions made by physicians leading to measures that are not compatible with an acute phase of CN.

## 2. Case Presentation

Our 24-year-old female patient was being treated for type I diabetes diagnosed at the age of 9 years. She had no other personal or family medical history. Her diabetes was complicated by sensitive neuropathy confirmed on an abnormal monofilament test and by severe retinopathy treated with laser pan-photocoagulation. She consulted in April 2020 for bilateral pre-malollary oedema that had been present from more than 4 months, with both joints presenting red swelling with local hyperthermia compared with the rest of the lower limb. However, this hyperthermia was almost the same in both ankles (Figure 1).

The young patient stated that she had consulted her general practitioner following the appearance of oedema on her feet. The doctor prescribed a venous Doppler echography to search for deep thrombophlebitis, although the examination did not reveal this type of anomaly. Later, after a new consultation, the patient followed the advice of her doctor to attend physiotherapy sessions and significantly increase the blood circulation in her lower limbs. The patient thus engaged in intense physical activity with a sports coach, jogging on a treadmill for more than 1½ hours, three times per week for more than 4 months. An X-ray of two feet taken after the first consultation with the general practitioner in April 2020 did not show any fractures.

As the aspect of the foot did not improve and oedemas developed, the patient then consulted with our specialist diabetic foot centre.

The patient was 162 cm tall and weighed 58 kg (body mass index 22.3 kg/m^2^). Her clinical history revealed HbA1c levels of 84.4 mmol/mol (9.8%), which was stable compared with the level recorded 3 months prior (9.7%). Other laboratory tests showed microalbuminuria (37 mg/day) with normal glomerular filtration rate (GFR 87 mL/min/1.73 m^2^). A bilateral foot MRI and computerised tomography scan (Figure 2) were performed, showing active bilateral CN in the talocalcaneal joint, with highly evolved bilateral osteolytic lesions of particular interest in the calcaneus and talus and signs of talonavicular arthropathy. Additionally, we observed multiple bone fragments and an effusion in the bone structure. Moreover, the patient reported no other medical problem or treatment aside from subcutaneous insulin therapy. The therapeutic option for this patient was complete offloading and the application of a plaster cast to unload the foot with the use of a wheelchair. A preventive dose of anticoagulant was prescribed. The follow-up was conducted in our diabetic foot centre with an MRI being performed 3 months after the start of offloading. It is important to point out that due to the oedema and signs of acute inflammation, the orthopaedist refused to perform surgery.

## 3. Discussion

CN is characterised by painful or painless bone and joint destruction in limbs that have lost sensory innervation with the clinical presentation occurring in two phases, namely, acute and chronic [1]. The incidence and prevalence of CN varies between 0.1% and 0.4% in people with diabetes [5,6]. The unilateral presentation of CN is much more common than bilateral involvement [7]. However, there is a relative risk of developing multifocal CN in 9% of people with CN [8].

The cause of this complication has not been completely elucidated. Several theories have been published, although the neuro-bone-inflammatory theory seems to be the most complete to explain CN. People with CN tend to have lower bone density in the lower limbs compared with neuropathic participants [9]. Studies using markers for bone formation and resorption highlighted increased osteoclastic activity compared with osteoblastic activity in the acute and chronic forms of CN [10,11]. In 2007, Jeffcoate [12] described CN as an increased inflammatory response to trauma, inducing increased bone lysis with the involvement of bone moulding factors, such as the receptor activator of the nuclear factor-B ligand, and its natural antagonist, osteoprotegerin.

Due to the potentially devastating consequences of CN, its rapid diagnosis and treatment are essential. Although surgical treatment is more likely to be used in the chronic phase of CN [13,14,15], the current treatment of CN in the acute phase consists of prolonged immobilisation with full offloading and a removable cast (e.g., Aircast^®^) [16,17]. The earlier offloading is started, the better the outcome [18]. Offloading gives the affected foot time to heal and thus prevent progressive damage and deformation of the bone structure. Offloading is maintained for as long as the foot shows signs of inflammation, with the average duration varying between 3 to 12 months in data sets [19] As treatment must be strictly adhered to over the long term to ensure success, patient compliance is an important and sensitive issue [20]. In addition to offloading, patients often require management for foot ulcers and possibly surgical reconstruction for disabling bone or joint deformity or instability. When the foot is in remission, extensive follow-up is required to provide the correct orthopaedic footwear and monitor for signs of reactivation, which would require further offloading. Risk factors for recurrence remain somewhat unexplored [21] and poorly identified. Non-detachment of CN in the active phase can lead to advanced bone necrosis and lead to the disappearance of the affected joint [22]. It is important to note that all clinical trials based on the use of drug therapies for the management of CN have shown promising results depending on the duration of the offloading, even if they sometimes revealed a reduction in its duration [23,24,25].

Our young patient probably consulted her general practitioner when CN was at stage 0 (oedema without fracture). Not only was her management of CN delayed, but her doctor’s recommendation to intensify her physical activity was in complete contradiction to the disease and probably aggravated it. The appearance of the fracture and the major deformation of the bone structure observed on the MRI seem related to this intensification of physical activity. The non-implementation of offloading was also a missed opportunity to reduce the impact of CN on the bone structure and limit its progression. Indeed, as the only validated treatment against bone destruction, offloading should be implemented as first-line management [16,17,18].

Our study not only sheds light on consequences of delays in the diagnosis of CN, known in the literature to be close to 86.9 days [26], but also provides a description of a chain of management errors with the implementation of care that probably induced the transition of CN from an insipid state to the level of deformity.

## Figures and Tables

**Figure 1 medicina-58-01763-f001:**
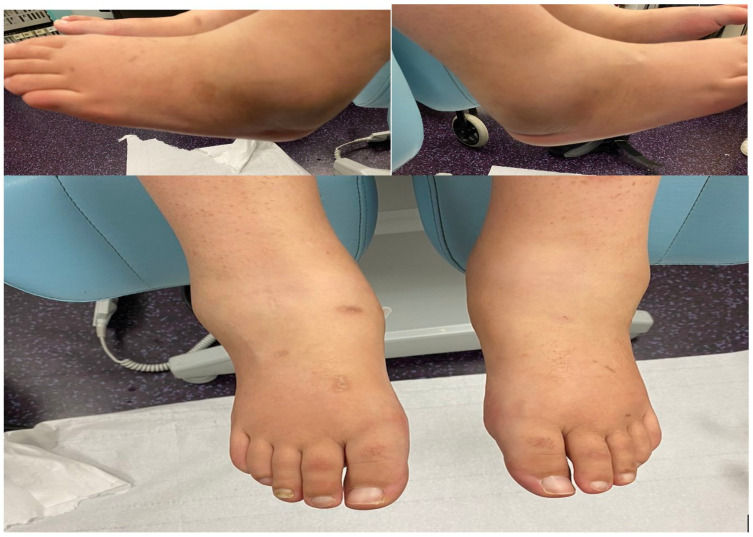
Clinical presentation of acute bilateral Charcot neuroarthropathy.

**Figure 2 medicina-58-01763-f002:**
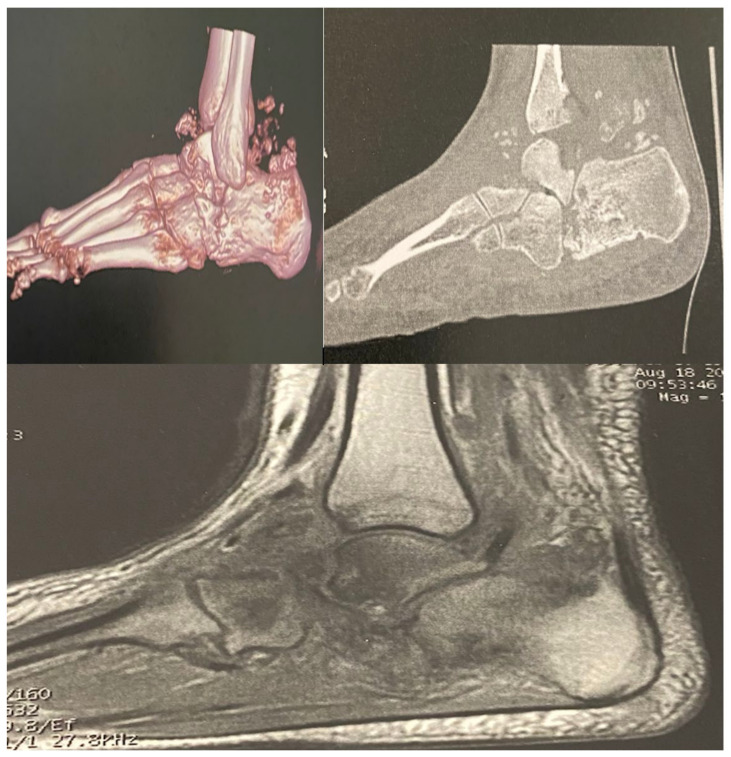
Computed tomography scan and magnetic resonance imaging showing the active presentation of bilateral Charcot neuroarthropathy.

## Data Availability

The data that support the findings of this study are available from the corresponding author.

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
