# Peer review of "Delayed Diagnosis of Bilateral Neuroarthropathy: Serious Impact on the Development of Charcot’s Foot, a Case Report"

_medicina, 2022, doi:10.3390/medicina58121763_

Round 1
Reviewer 1 Report
Title
Delayed diagnosis of bilateral presentation of neuroarthropa-thy, a serious impact on the development of charcot's foot, a case report
The authors discussed a case of charcot's arthropathy that deteriorated due to delayed diagnosis. Some important point need to be addressed .
|
Line |
Manuscript |
Comment |
|
14 |
Results: |
The case should be presented first |
|
14 |
the case of a lady |
English editing is mandatory |
|
14 |
the case of a lady |
The authors should described the age of the case |
|
15 |
in only 4 months |
English editing is mandatory |
|
16 |
but it was also multifocal, affecting both tarsal 16 joints. |
The English writing needs revision |
|
18 |
none of the physicians consulted made a suspicion or a diagnosis of CN |
The sentence is too hard to be understood |
|
Abstract |
|
The abstract is reluctant and the description of the case is not so interesting The abstract should be written in attractive mannar to attract the readers |
|
Conclusion 19-21 |
|
The conclusion not reflected by the case description |
|
29
|
lysis, The clinical presentation of 29 CN :The |
There is something wrong with the use of capital and small letters and the use of punctuation |
|
35 |
Eichenholtz [2]. (assessment |
Here again |
|
39 |
where magnetic 39 resonance imaging |
In appropriate insertion of abbreviation and their meanings |
|
|
|
The introduction should also more interesting and attractable to the authors |
|
Patients and Methods |
|
I think case presentation will be more appropriate as a title than patients and methods |
|
Our patient was aged 24 years |
|
Not a scientific writing |
|
oedema present from 55 more than 4 months |
|
The sentence structure is not appropriate |
|
56 |
both joints presenting an inflammatory aspect |
What is meant by inflammatory aspect |
|
68 |
The clinical 68 history of the patient shows the presence of an HbA1c level of |
This is not a clinical history .This is laboratory data |
|
Case presentation |
|
Should be mentioned in a chronological manner |
Author Response
Dear Reviewer: many thanks for your comments, attached are the responses to your comments
Kindly regards
Dured Dardari

Reviewer 2 Report
Paper does not give us novelties. Nevertheless, the message you want to highlight is very important as to date the Charcot joint is often unrecognized and undiagnosed.
You should rearrange the description of the case improving the clinical features description and highlighting the moment in which somebody has had a wrong diagnosys, explaining why and how he should have done.
Plesase, be careful of the fluidity of sentences and the use of punctuation (some sentences need to be rewritten)
Author Response
Dear Reviewer
many thanks for your comments we have reviewed our entire manuscript and we hope that the new version will meet your expectations
Kindly regards
Dured Dardari
Reviewer 3 Report
Many thanks to the authors for having presented a so interesting Case Report about “Delayed diagnosis of bilateral presentation of neuroarthropathy, a serious impact on the development of Charcot’s foot, a case report”. Before resubmitting the revision version of the article, please read the editorial rules carefully, and check other editorial aspects, such as: text alignment (lacking), text justification at the head (lacking), etc. There are too many typing errors! The language is good enough. Please correct the title with italic, uppercase, letter for Charcot foot.
Abstract
The abstract is well structured, and it contains the main information of the study.
Background
Although the introduction is quite well structured it is too short and shallow. Please make the introduction wider.
About operative management for Charcot foot please add minimally invasive techniques quoting the most recent reports presented in literature:
· Distal Metatarsal Osteotomies for Chronic Plantar Diabetic Foot Ulcers. Foot Ankle Clin. 2022 Sep;27(3):545-566. doi: 10.1016/j.fcl.2022.02.003. Epub 2022 Aug 6. PMID: 36096551
· Minimally invasive metatarsal osteotomies (Mimos) for the treatment of plantar diabetic forefoot ulcers (pdfus): A systematic review and meta‐analysis with meta‐regressions. Applied Sciences (Switzerland), 2021, 11(20), 9628
Patients and Methods
This section contains enough information.
When was the sensitive neuropathy diagnosed? Please add images of RMI performed during follow up.
Discussion
The length and content of the discussion communicates the main information of the paper.
However, the case presented is not discussed adequately with data provide in literature, especially in relation to possible surgical techniques.
Lines 123-24: In addition to offloading, patients will often require management for the development of foot ulcers and possibly surgical reconstruction for disabling bone or joint deformity or instability.
Please discuss possible future surgical options for this case and quote respectively For bone reconstruction after immobilization in cast or boot:
· Minimally Invasive Surgery for Tibiotalocalcaneal Arthrodesis Using a Retrograde Intramedullary Nail: Preliminary Results of an Innovative Modified Technique. J Foot Ankle Surg. 2016 Nov-Dec;55(6):1130-1138. doi: 10.1053/j.jfas.2016.06.002. Epub 2016 Aug 11.
References
The references are not up to date. Hence, delate those before 2010 if not essential, replacing them with newer ones and integrate them as suggested previously.
Figures
The quality of figures is low. It would be advisable to add the clinical and radiographic images taken during follow-up.
Author Response
Dear reviewer many thanks for your comments we have reviewed our entire manuscript and we hope that the new version will meet your expectations
Abstract
The abstract is well structured, and it contains the main information of the study.
Modification done
Although the introduction is quite well structured it is too short and shallow. Please make the introduction wider.
About operative management for Charcot foot please add minimally invasive techniques quoting the most recent reports presented in literature:
- Distal Metatarsal Osteotomies for Chronic Plantar Diabetic Foot Ulcers. Foot Ankle Clin. 2022 Sep;27(3):545-566. doi: 10.1016/j.fcl.2022.02.003. Epub 2022 Aug 6. PMID: 36096551
- Minimally invasive metatarsal osteotomies (Mimos) for the treatment of plantar diabetic forefoot ulcers (pdfus): A systematic review and meta‐analysis with meta‐regressions. Applied Sciences (Switzerland), 2021, 11(20), 9628
Many thanks for your comments : Modification done
Round 2
Reviewer 1 Report
Dear author,
The reply to reviewer comments should be written in a scientific manner to indicate what has already been done. The paper as a whole is full of multiple caveats and not well written.
Reviewer 3 Report
The new references added must also reported the names of the authors. Please provide them.